# Exploring Interrelationships between Colour, Composition, and Coagulation Traits of Milk from Cows, Goats, and Sheep

**DOI:** 10.3390/foods13040610

**Published:** 2024-02-17

**Authors:** Ana Garzón, José M. Perea, Elena Angón, Eoin G. Ryan, Orla M. Keane, Javier Caballero-Villalobos

**Affiliations:** 1Department of Animal Production, University of Córdoba, 14071 Córdoba, Spain; pa1gasia@uco.es (A.G.); jmperea@uco.es (J.M.P.); eangon@uco.es (E.A.); 2Section of Herd Health and Animal Husbandry, School of Veterinary Medicine, University College Dublin, D04 V1W8 Belfield, Ireland; eoin.g.ryan@ucd.ie; 3Teagasc Animal & Bioscience Research Department, Grange, C15 PW93 Dunsany, Co. Meath, Ireland; orla.keane@teagasc.ie

**Keywords:** milk, dairy, cow, goat, sheep, coagulation, colour values, technological quality

## Abstract

This study explores the interrelationships between the composition, coagulation, and colour of sheep, goat, and cow milk to identify their similarities and differences and to assess whether the relationships between the variables are common to all species or whether they emerge from species-specific relationships. For this purpose, 2400 individual milk samples were analysed. The differences and similarities between the species were determined using discriminant analysis and cluster analysis. The results show a clear differentiation between species. Sheep milk stands out for its cheesemaking capacity and shows similarities with goat milk in composition and coagulation. Nonetheless, colorimetry highlights a greater similarity between sheep and cow milk. Composition and colorimetry were more discriminating than coagulation, and the variables that differed the most were fat, protein, curd yield, lightness, and red–green balance. Using canonical correlation, the interrelationships between the different sets of variables were explored, revealing patterns of common variation and species-specific relationships. Colorimetric variables were closely related to milk solids in all species, while in sheep milk, an inverse relationship with lactose was also identified. Furthermore, a strong relationship was revealed for all species between colour and curd yield. This could be modelled and applied to estimate the technological value of milk, proving colorimetry as a useful tool for the dairy industry.

## 1. Introduction

Breeding programs have traditionally focused their efforts on the quality of bovine milk. However, when extending these programs to small ruminants, notable differences have emerged in milk composition and milk coagulation properties (MCP) between the species of interest [1,2]. Milk coagulation performance acquires great relevance, especially in some species such as sheep and goats, since almost the totality of their milk is used for cheesemaking [3]. Thus, deficiencies in this process can greatly affect production, potentially leading to massive economic losses in the dairy industry [4].

Recent research has delved into the analysis of milk traits and the coagulation process in different ruminant species, identifying common patterns and singularities that influence its cheesemaking aptitude [5,6,7,8]. A comparative study analysed milk composition and MCP in six species, revealing that sheep milk has superior coagulation properties compared to bovine and goat milk [9]. On the other hand, bovine milk showed slower coagulation, while goat milk was characterised by a rapid loss of curd firmness after reaching its maximum peak. More recently, other authors explored variations in milk from sheep, goats, and cows regarding its processing, gelation, and seasonal factors, providing new perspectives for the adjustment of dairy products to each species [10].

Milk colour emerges as a potential indicator of quality in terms of composition and technological performance. In the case of bovine milk, many authors have established a correlation between chromaticity and traits such as fat content and other hygienic–sanitary factors [11,12,13]. Meanwhile, research on sheep milk suggests that colour indexes could be an effective tool to predict composition and coagulation parameters [14]. However, the direct relationship between the colour of milk and its coagulation behaviour has not yet been studied in depth. The chromatic differences observed between the milk of various species point out that colour may be associated with the unique characteristics of each type of milk [15]. These findings stress the importance of engaging in comparative analyses that identify differences and similarities between species of interest and help understand the existence of distinctive traits that have specific effects on the coagulation process and its performance. This knowledge is, therefore, deemed key to optimise the quality and efficiency of dairy manufacturing processes, particularly in the cheese industry.

Thus, this study has two main goals: Firstly, to identify differences and similarities between milk from cows, sheep, and goats, regarding composition, coagulation, and colour. Secondly, to explore whether relationships between these variables are common to the three species or whether species-specific relationships emerge. These objectives aim to achieve a deeper understanding of both the general attributes of milk in the species of interest and the particular factors that influence quality in cheese production.

## 2. Materials and Methods

### 2.1. Dataset

The study was conducted with 2400 individual milk samples from three domestic ruminant species: sheep, goats, and cows. Data were collected from the extensive historic database of milk samples analysed at the Dairy Laboratory of the University of Córdoba (Spain) over the past ten years, and the *RANDBETWEEN* function of MS Excel (Microsoft Corp., Redmond, WA, USA) was used to randomly select 800 individual samples from each species. This approach was adopted to mitigate any potential biases and to capture a broad spectrum of variability within species.

### 2.2. Laboratory Analysis

All the samples included in this research were collected in similar conditions during morning milkings and kept in cold storage at 4 °C in hermetically sealed containers until analysis, which was performed at UCO Dairy Laboratory (Department of Animal Production, University of Córdoba, Spain).

Native pH (pH) of milk was measured using a Crison Basic20 pH meter (Crison Instruments S.A., Barcelona, Spain), and milk major components—fat content (FAT), crude protein (CP), and lactose (LAC)—were determined by mid-infrared spectroscopy on a MilkoScan FT120 (Foss Electric, Hillerød, Denmark). Milk coagulation properties (MCP) were monitored at 32 °C employing a classic Formagraph viscometer (Foss Electric) [16], obtaining values for rennet clotting time (RCT), curd firming time (k_20_) and curd firmness at 60 min (A_60_). Curd yield (CY) was expressed as g/10 mL of milk after cutting the fresh curds with a spatula and draining by centrifugation at 2800× *g* and 37 °C for 30 min, and dry curd yield (DCY) was calculated after desiccating the curds in a drying oven at 100 °C for 24 h, and expressed as a percentage of CY [8,17]. Colour indexes of milk were expressed as three variables, using the CIELAB colour space [18]. For this purpose, lightness (L*), green–red balance (a*), and blue–yellow balance (b*) were measured using a PCE-CSM2 Colour Meter (PCE Instruments Ltd., Southampton, UK) placed directly over a capsule containing 2 mL of the sample [14].

### 2.3. Statistical Analysis

Preliminary data testing was conducted to identify and discard outliers before proceeding with further analysis. Given the diverse measurement units of the data, standardisation was performed to achieve a zero mean and a unit standard deviation. The comprehensive descriptive characteristics of the variables under research are presented in Table 1.

Multivariate analysis was utilised to address the primary inquiries of this study: (1) Are composition, colorimetric variables, and coagulation properties similar among the species of interest? (2) Do the same interrelationships exist among the variables defining colour, composition, and coagulation process in the three species of interest?

The first research question was addressed through two multivariate techniques. Firstly, Canonical Discriminant Analysis (CDA) was applied, providing insights into the stepwise and overall similarities among groups of variables—colorimetry, composition, and coagulation properties [19]. Canonical discriminant analysis was chosen for its effectiveness in reducing dimensionality and optimising the separation between groups, offering a clear interpretation of the contributions of variables to discrimination, especially in large datasets. The analysis was conducted separately for each set of variables and for the entire group. Discriminative power was evaluated through the significance test of Wilks’ lambda value, and prediction capacity was tested using absolute animal assignment to pre-assigned groups. Mahalanobis distances were employed to determine group distances, and stepwise discriminant analysis assessed the discrimination ability of variables. CDA yielded a graphical representation of observations in the space formed using the first two grouping variables, visually confirming the existence of groups among variables [20].

The second method involved clustering based on Euclidean distances among groups calculated with individual Mahalanobis distances [21]. This analysis elucidated concrete relationships among discriminated groups. Cluster analysis, suitable for simple representation and quantification of relationships among groups [22], complemented discriminant analysis by explaining associations between data. The results are provided as individual plots representing the determined clusters and their linkage points.

The second research question was addressed by Canonical Correlation Analysis (CCA) among variable groups within each block (colorimetry, composition, and coagulation). CCA, a multivariate statistical modelling technique for studying interrelationships among groups of variables, complements discriminant analysis [23]. Canonical correlations were analysed by paired groups of variables, obtaining values and significance through Chi-square tests with successive roots removed [24]. Canonical coefficients of determinations and indicators of robustness, such as calculated variance and total redundancy, were presented for each set of variables [25]. Interrelationships within the species of interest were scrutinised and all statistical analyses were executed using XLSTAT v.19.4 (Addinsoft, New York, NY, USA).

## 3. Results

### 3.1. Differentiation of Milk from Dairy Species

Table 2 presents the outcomes of the canonical discriminant analysis using the variables related to milk composition, coagulation, colorimetry, and the whole set of variables. The table emphasises the variables selected by stepwise discriminant analysis that showed higher discriminatory ability among species. The most discriminative composition variables were FAT and CP. LAC and pH also yielded significant results but exhibited a lower discriminatory power. In terms of coagulation properties, CY emerged as the variable with the highest discriminatory power. All three evaluated colorimetric variables were significant, with L* and a* being the most discriminatory between species.

Sheep milk stood out for its higher nutrient content, followed by goat and cow milk. Goat milk showed higher FAT than cow milk, but lower protein content. Regarding MCP, sheep milk was characterised by lower k_20_, A_60_, and CY, although DCY was similar to that of goat milk. Cow milk displayed the lowest yields (both CY and DCY), coagulated slower, and reached an intermediate curd firmness. The colorimetric variables outlined sheep milk as having the highest L* and b*, while cow milk was characterised by lower L*, a*, and b* indexes. Goat milk showed L* levels similar to sheep milk, with the highest a* values.

Canonical discriminant analysis was applied to the selected variables in each of the four sets (Table 3). In all cases, the extracted canonical functions significantly discriminated among the three species (*p* < 0.001, Hotelling’s T^2^ test). The F-statistics revealed a higher discriminating ability for the variables related to milk composition and colorimetry. This is also evident in Table 4, which presents the values for Mahalanobis distance between the three species in each set of variables. All pairwise distances were significant. This is further illustrated in the graphic bidimensional representation of the results (Figure 1).

Cluster analysis supported these findings, as the Euclidean distances obtained resulted in clear divisions between species, that are easily noticeable (Figure 2). Cluster analysis showed the highest similarity between sheep and goat milk and the greatest dissimilarity for cow milk, except for colorimetric variables, where similarities were observed between sheep and cow milk, while goat milk differed from both.

Discriminant analysis for the whole set of variables correctly classified 97.8% of the samples into their original species (Table 5). The model based on composition variables classified 91.9% of the samples correctly, the model based on coagulation properties correctly classified 82.2%, and the model based on colorimetric variables correctly classified 88.8%.

The classification error for the positive predictions ranged from 2.3% in the model based on the whole set of variables to 17% in the model based on the coagulation properties. For the negative predictions, the classification error varied between 2.2% for the model based on the whole set of variables and 17% for the model based on the coagulation properties.

### 3.2. Relationship among the Groups of Variables

Through the CCA models, we aimed to address whether similar relationships are established among the variables defining colour, composition, and coagulation processes, or conversely, if each dairy system establishes specific interrelationships associated with the species. Nine CCA models were developed and their general characteristics are outlined in Table 6. High and statistically significant canonical correlations were found within the overall framework of all the analysed systems, both among the colorimetric variables and between the composition and the coagulation properties in each of the analysed species. The correlation structure is depicted in Table 7.

When the composition was analysed with MCP, there was enough evidence to confirm that the composition variables were strongly correlated with the coagulation properties in all the evaluated species. However, these relationships were species-specific. In the milk from goats, the first canonical component exhibited a strong and positive relationship between FAT and CP with CY (Table 7). In the milk from cows and ewes, the first component also included a positive relationship with A_60_ and DCY. The second component showed some similarities in bovine and ovine milk, but considerable differences from caprine milk. In the milk from goats, the second component positively related A_60_ to CP and negatively to FAT. In the milk from cows and ewes, a strong and positive relationship was established between k_20_ and pH. In the milk from cows, RCT also appeared.

The relationship between the colorimetric variables and the composition was strong in all the analysed species (Table 6). The first canonical component positively related FAT and CP to all the colorimetric variables in caprine milk, with L* and b* in bovine milk, and with a* and b* in ovine milk, also including an inverse relationship with LAC (Table 7). The second canonical component relationships were more specific. In the bovine milk, a positive relationship was established between a* and pH, and a negative relationship with LAC. In the goat milk, a negative relationship was established between a* and CP. In the ovine milk, a negative relationship was established between pH and L*.

The relationship between the colorimetric variables and the coagulation properties was also strong in all the analysed species (Table 6). The first canonical component was quite similar among dairy species and showed a positive relationship between CY and colorimetric variables L*, b*, and a*, the latter only in ovine and caprine milk (Table 7). The second canonical component was more species-specific. In the ovine milk, an inverse relationship was established between k_20_ and RCT with L*. In the bovine milk, a positive relationship was established between a* with k_20_ and RCT, and a negative relationship with DCY. In the caprine milk, a positive relationship was established between L* and A_60_, and a negative relationship with k_20_ and RCT.

## 4. Discussion

Discriminant analysis and cluster analysis have revealed both significant differences and similarities between milk from the three studied ruminant species. There is a clear differentiation between cow and sheep milk, while goat milk, although clearly differentiated, shows some similarity to sheep milk. This is clearly represented in Figure 1 and Table 5, where despite the low incidence of classification errors, these are more frequent among goat and sheep milk samples, and to a lesser extent, among cow and goat milk.

Studies comparing the composition and cheesemaking aptitude of milk from different species using a similar approach are scarce [9,26,27,28] and, to the authors’ knowledge, there are no available studies that compare relationships between composition, colour, and coagulation properties in milk from domestic ruminants.

The milk composition values reported in this study are within the usual ranges previously described by other authors [26,27,28,29,30]. The variability in milk composition between the different species is associated with a set of genetic, dietary, and metabolic factors inherent to its production [9,29]. Furthermore, other important variations within species are due to factors such as stage of lactation, herd/flock, nutrition, climate, breed, or season of the year [7,31,32].

The greater nutritional richness and similarity between sheep and goat milk can be attributed, to some extent, to the evolutionary proximity of both species, which not only involves physiological resemblances but also similar zootechnical and husbandry practices [33,34]. These strategies include genetic selection schemes involving different approaches from those used in cattle [1,35] since practically, the totality of milk production in small ruminants is used for the manufacture of cheese and other dairy products [36,37].

The variability in MCP between the different species was lower and highlights more marked similarities between sheep and goat milk, which is in accordance with previous research [9,34,38]. CY and, to a lesser extent, DCY, have proved to be the most discriminating variables between the species. When analysing in conjunction with the canonical correlation models, common patterns and species-specific traits emerge.

In the three species considered, an expected strong link between CY, FAT, and CP can be observed [7,8,38], which in sheep and cows also relates to DCY and A_60_. These outcomes suggest that bovine and sheep milk follow a more similar coagulation pattern in which a denser and firmer clot favours the retention of fat and other solids, resulting in a more enriched curd that leads to higher yields. Hence, different curd yields obtained from bovine and sheep milk can be attributed to differences in nutrient content and curd firmness, which agrees with other previous studies [6,9]. However, goat milk seems to obtain intermediate curd yields with lower firmness, and a direct relationship between A_60_ and CY is not evident. According to some authors [39], this is the main reason for the lower cheese yield of goat milk. However, other studies [38,40] suggest that, ultimately, the variations in the nutrient content of milk are responsible for the differences in the curd yields in goat milk.

Colour variables have proven to be highly effective in discriminating between the three types of milk, providing better outcomes than the coagulation properties. However, they do not reach the level of discrimination obtained by directly comparing milk composition between species. Canonical correlation models show that the colour indexes are strongly related to FAT and CP and, in sheep milk, also to LAC. With regard to the study’s limitations, there are obviously other components in milk that have not been measured in this study that may influence its colour [13,41], which should be considered in future studies. In addition, it would be interesting to further explore the relationship between the colour and the taste of the milk since both are reported to influence consumers’ perception of dairy products [15,42]. From the perspective of the cheese industry, the canonical correlation found between milk colour and its curdling performance acquires particular importance, since this seems to be a relationship common to the three species of interest. These findings open the possibility of optimising cheese production processes, allowing producers and cheesemakers to carry out quick and efficient assessments of milk based solely on colour, a parameter that has so far been underestimated as an indicator of technological quality.

## 5. Conclusions

The significant differences between cow, sheep, and goat milk are well known, and the present study highlights the particularities in composition, colour, and coagulation properties, proving that sheep milk stands out for its superior cheesemaking capacity. Sheep and goat milk show similarities in their composition and coagulation performance, while colorimetry suggests a greater similitude between sheep and cow milk. However, composition, coagulation, and colour are interrelated through both common patterns of variation and species-specific relationships. There is a strong positive relationship common to all three species between major milk solids (fat and protein content) and curd yield, while in sheep and cow milk, this relationship also extends to curd firmness and the dry extract of the curd. Colour variables were closely related to fat and protein content in all species and, in sheep milk, an inverse relationship with lactose concentration was also identified. Furthermore, a strong connection between colour indexes and curd yield was evidenced in the three species. This link could be modelled and applied to real situations to estimate the technological value of milk, thus making colorimetry a potentially valuable tool for the dairy industry in optimising cheese production.

## Figures and Tables

**Figure 1 foods-13-00610-f001:**
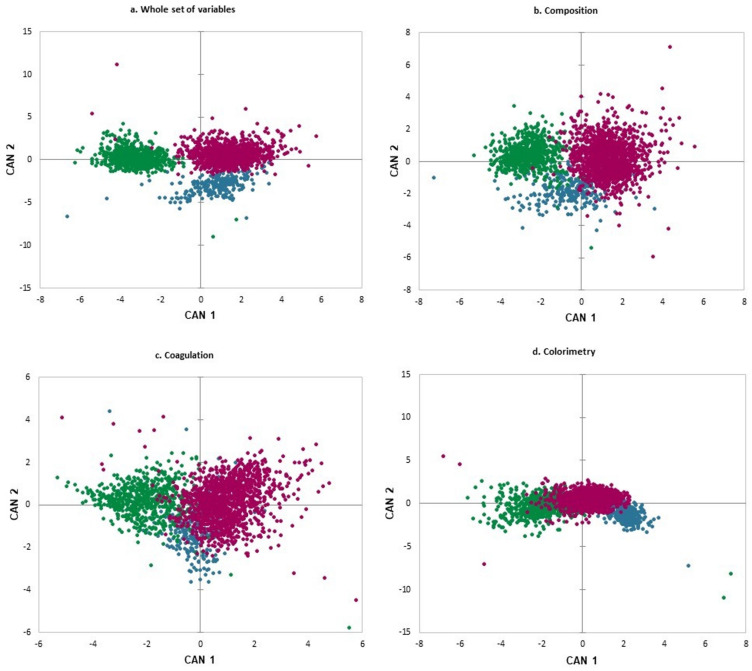
Graphical representation of the results from canonical discriminant analysis defined by the axes of the 2 first canonical variables (CAN 1 and CAN 2) for (**a**) the whole set of variables; (**b**) composition variables; (**c**) coagulation variables; (**d**) colorimetric variables. (●) Goat, (●) Cow, (●) Sheep.

**Figure 2 foods-13-00610-f002:**
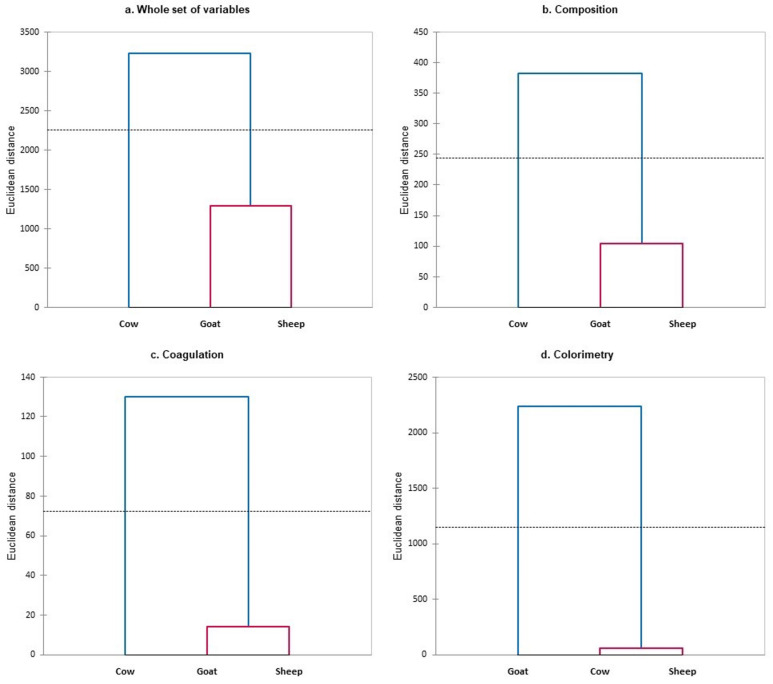
Representation of the results from cluster analysis for (**a**) the whole sets of variables; (**b**) composition variables; (**c**) coagulation variables; (**d**) colorimetric variables. Different coloured lines indicate different groups according to the Hartigan index (represented by a dashed line).

**Table 1 foods-13-00610-t001:** Descriptive characteristics of the variables under research (*n* = 2400).

Variable	Description	Unit/Range
FAT	Fat content	%
CP	Crude protein	%
LAC	Lactose content	%
pH	pH	−log[H^+^]
L*	Lightness	0, 100
a*	red/green balance	−60, +60
b*	blue/yellow balance	−60, +60
RCT	Rennet clothing time	min
k_20_	Curd firming time	min
A_60_	Curd firmness at 60 min	mm
CY	Curd yield	g/10 mL of milk
DCY	Dry curd yield	%

**Table 2 foods-13-00610-t002:** Results of canonical discriminant analysis with all variables measured for composition, coagulation, colorimetry, and whole set of variables (variables with the greatest discriminant ability are shown in bold), using Student–Newman–Keuls (SNK) test, and correlations of each variable with the canonical variables (CAN).

Variable ^1,^*	Cow	Goat	Sheep	Wilks’ λ	F-Value	*p*-Value	R^2 2^	CAN_1_ ^3^	CAN_2_ ^3^
All variables									
**FAT**	**2.44 ± 1.12 ^c^**	**5.56 ± 1.23 ^b^**	**6.54 ± 1.81 ^a^**	**0.429**	**1518.51**	**<0.001**	**0.867**	**0.835**	**0.072**
**CP**	**4.18± 0.47 ^b^**	**3.98 ± 0.54 ^c^**	**5.59 ± 0.79 ^a^**	**0.474**	**1264.47**	**<0.001**	**0.775**	**0.632**	**0.603**
LAC	4.62 ± 0.26 ^c^	4.86 ± 0.45 ^b^	4.95 ± 0.36 ^a^	0.850	201.44	<0.001	0.283	0.427	0.052
pH	6.70 ± 0.08 ^a^	6.66 ± 0.13 ^b^	6.61 ± 0.14 ^c^	0.898	130.12	<0.001	0.423	−0.343	−0.111
RCT	20.48 ± 6.00	20.07 ± 7.38	19.88 ± 10.03	0.999	1.07	0.343	0.473	−0.034	−0.005
k_20_	9.39 ± 6.01 ^a^	6.06 ± 3.95 ^b^	3.53 ± 3.09 ^c^	0.730	420.56	<0.001	0.518	−0.562	−0.145
A_60_	30.94 ± 9.44 ^b^	25.09 ± 9.65 ^c^	38.69 ± 10.99 ^a^	0.817	255.28	<0.001	0.384	0.280	0.464
**CY**	**16.64 ± 3.72 ^c^**	**20.59 ± 4.78 ^b^**	**26.76 ± 5.77 ^a^**	**0.558**	**903.79**	**<0.001**	**0.879**	**0.695**	**0.295**
DCY	35.02 ± 4.09 ^b^	42.17 ± 6.08 ^a^	42.37 ± 5.43 ^a^	0.694	502.89	<0.001	0.536	0.611	−0.059
**L***	**78.27 ± 2.87 ^c^**	**83.47 ± 1.28 ^b^**	**83.61 ± 2.21 ^a^**	**0.483**	**1221.35**	**<0.001**	**0.736**	**0.794**	**−0.077**
**a***	**−4.13 ± 1.34 ^c^**	**−1.12 ± 0.51 ^a^**	**−2.46 ± 0.71 ^b^**	**0.483**	**1217.35**	**<0.001**	**0.707**	**0.679**	**−0.505**
b*	2.52 ± 3.17 ^c^	3.29 ± 1.22 ^b^	4.49 ± 1.93 ^a^	0.870	169.63	<0.001	0.572	0.377	0.159
*Composition*									
**FAT**	**2.44 ± 1.12 ^c^**	**5.56 ± 1.23 ^b^**	**6.54 ± 1.81 ^a^**	**0.429**	**1518.51**	**<0.001**	**0.491**	**0.855**	**−0.266**
**CP**	**4.18± 0.47 ^b^**	**3.98 ± 0.54 ^c^**	**5.59 ± 0.79 ^a^**	**0.474**	**1264.47**	**<0.001**	**0.482**	**0.770**	**0.512**
**LAC**	4.62 ± 0.26 ^c^	4.86 ± 0.45 ^b^	4.95 ± 0.36 ^a^	0.850	201.44	<0.001	0.021	0.441	−0.116
**pH**	6.70 ± 0.08 ^a^	6.66 ± 0.13 ^b^	6.61 ± 0.14 ^c^	0.898	130.12	<0.001	0.077	−0.370	0.004
*Coagulation*									
**RCT**	20.48 ± 6.00	20.07 ± 7.38	19.88 ± 10.03	0.999	1.07	0.343	0.286	−0.038	0.004
**k20**	9.39 ± 6.01 ^a^	6.06 ± 3.95 ^b^	3.53 ± 3.09 ^c^	0.730	420.56	<0.001	0.467	−0.647	−0.048
**A60**	30.94 ± 9.44 ^b^	25.09 ± 9.65 ^c^	38.69 ± 10.99 ^a^	0.817	255.28	<0.001	0.223	0.385	0.797
**CY**	**16.64 ± 3.72 ^c^**	**20.59 ± 4.78 ^b^**	**26.76 ± 5.77 ^a^**	**0.558**	**903.79**	**<0.001**	**0.307**	**0.819**	**0.289**
**DCY**	35.02 ± 4.09 ^b^	42.17 ± 6.08 ^a^	42.37 ± 5.43 ^a^	0.694	502.89	<0.001	0.132	0.668	−0.374
*Colorimetry*									
**L***	**78.27 ± 2.87 ^c^**	**83.47 ± 1.28 ^b^**	**83.61 ± 2.21 ^a^**	**0.483**	**1221.35**	**<0.001**	**0.605**	**0.821**	**0.564**
**a***	**−4.13 ± 1.34 ^c^**	**−1.12 ± 0.51 ^a^**	**−2.46 ± 0.71 ^b^**	**0.483**	**1217.35**	**<0.001**	**0.597**	**0.903**	**−0.047**
**b***	2.52 ± 3.17 ^c^	3.29 ± 1.22 ^b^	4.49 ± 1.93 ^a^	0.870	169.63	<0.001	0.409	0.300	0.504

^1^ Means without a common superscript (a–c) are statistically different (*p* < 0.05) by SNK test. ^2^ R^2^ = 1—tolerance. ^3^ Correlation of each variable with the canonical variable. * FAT = fat content; CP = crude protein; LAC = lactose content; RCT = rennet clotting time; k_20_ = curd firming time; A_60_ = curd firmness at 60 min; CY = curd yield; DCY = dry curd yield; L* = lightness; a* = red/green balance; b* = blue/yellow balance.

**Table 3 foods-13-00610-t003:** Results from the canonical discriminant analysis for composition, coagulation, colorimetric variables, and whole set of variables.

Model	Variables in Model, No.	Wilks’ λ	F-Value	*p*-Value
Whole set	12	0.083	468.30	<0.001
Composition	4	0.172	802.49	<0.001
Coagulation	5	0.308	364.78	<0.001
Colorimetry	3	0.261	725.89	<0.001

**Table 4 foods-13-00610-t004:** Mahalanobis distances between the studied species for milk coagulation variables (under the diagonal), milk composition variables (under the diagonal, in parentheses), colorimetric variables (above the diagonal), and whole set of variables (above the diagonal, in parentheses).

Dairy System	Goat	Cow	Sheep
Goat		13.51 (37.04)	7.87 (15.70)
Cow	6.58 (10.69)		7.64 (25.61)
Sheep	3.13 (9.41)	11.98 (21.91)	

All distances are significant at *p* < 0.001.

**Table 5 foods-13-00610-t005:** Discriminant analysis showing the percentage of samples correctly classified by species using four models—all variables; composition; coagulation; and colorimetry.

Model	Goat	Cow	Sheep
All variables			
Goat	94.72	1.51	3.77
Cow	0.15	98.66	1.19
Sheep	1.49	0.59	97.92
Error level	0.08	0.02	0.01
Priors	0.33	0.33	0.33
Composition			
Goat	87.92	3.02	9.06
Cow	3.86	94.95	1.19
Sheep	8.25	0.59	91.16
Error level	0.37	0.02	0.01
Priors	0.33	0.33	0.33
Coagulation			
Goat	76.98	6.04	16.98
Cow	10.57	86.90	2.53
Sheep	17.25	1.78	80.97
Error level	0.60	0.06	0.05
Priors	0.33	0.33	0.33
Colorimetry			
Goat	96.23	0.38	3.40
Cow	0.00	85.88	14.12
Sheep	3.64	7.50	88.86
Error level	0.16	0.15	0.08
Priors	0.33	0.33	0.33

**Table 6 foods-13-00610-t006:** Results of the canonical correlation analysis for the relationships between composition, coagulation, and colorimetry.

Root	Eigenvalue	Canonical Correlation	Cumulative Variability (%)	Lambda	F-Value	*p*-Value
Composition—coagulation model for cow						
F1	0.721	0.849	64.17	0.175	29.32	<0.001
F2	0.299	0.547	90.77	0.629	10.84	<0.001
F3	0.089	0.298	98.77	0.898	4.77	<0.001
Composition—coagulation model for goat						
F1	0.801	0.895	64.43	0.121	97.99	<0.001
F2	0.271	0.521	86.27	0.606	30.52	<0.001
F3	0.155	0.394	98.77	0.832	21.37	<0.001
Composition—coagulation model for sheep						
F1	0.797	0.893	61.41	0.110	209.99	<0.001
F2	0.402	0.634	92.39	0.540	77.43	<0.001
F3	0.087	0.295	99.09	0.902	23.52	<0.001
Composition—colorimetric model for cow						
F1	0.476	0.690	78.07	0.454	19.77	<0.001
F2	0.123	0.350	98.16	0.868	6.36	<0.001
Composition—colorimetric model for goat						
F1	0.592	0.769	83.33	0.361	68.94	<0.001
F2	0.088	0.297	95.75	0.884	14.09	<0.001
Composition—colorimetric model for sheep						
F1	0.516	0.719	74.37	0.401	121.71	<0.001
F2	0.104	0.323	89.35	0.830	43.71	<0.001
Coagulation—colorimetric model for cow						
F1	0.387	0.622	84.75	0.571	10.65	<0.001
F2	0.049	0.221	95.48	0.931	2.33	0.018
Coagulation—colorimetric model for goat						
F1	0.577	0.760	68.91	0.319	62.58	<0.001
F2	0.166	0.407	88.69	0.755	25.05	<0.001
Coagulation—colorimetric model for sheep						
F1	0.386	0.621	85.22	0.574	54.89	<0.001
F2	0.044	0.210	94.98	0.934	11.61	<0.001

**Table 7 foods-13-00610-t007:** Correlation coefficients between the variables and the canonical variables included in the canonical correlation analysis for each model and species of interest (variables with the highest correlation coefficient are in bold).

Variable *	Canonical Component					
	Cow		Goat		Sheep	
	F_1_	F_2_	F_1_	F_2_	F_1_	F_2_
Composition—coagulation models						
pH	−0.324	**0.823**	−0.150	−0.104	0.217	**0.890**
FAT	**−0.829**	−0.359	**0.809**	**0.569**	**−0.954**	−0.127
CP	**−0.800**	−0.112	**0.825**	**−0.519**	**−0.864**	0.184
LAC	−0.119	−0.455	−0.076	−0.126	−0.087	−0.301
RCT	−0.449	**0.661**	−0.078	−0.322	−0.117	**0.898**
k20	0.151	**0.702**	−0.397	0.409	0.499	0.478
A60	**−0.633**	0.251	0.234	**−0.563**	**−0.627**	0.161
CY	**−0.952**	−0.157	**0.968**	0.010	**−0.948**	0.103
DCY	**−0.561**	−0.231	0.130	−0.346	**−0.555**	−0.446
Composition—colorimetric models						
pH	−0.091	**0.677**	−0.199	0.272	−0.099	**−0.774**
FAT	**−0.865**	0.260	**0.943**	0.257	**0.899**	0.165
CP	**−0.796**	−0.154	**0.618**	**−0.755**	**0.823**	−0.228
LAC	−0.337	**−0.671**	−0.360	−0.239	**−0.793**	0.350
L*	**−0.920**	−0.166	**0.653**	−0.136	0.472	**0.840**
a*	−0.055	**0.869**	**0.746**	**0.516**	**0.748**	0.371
b*	**−0.791**	0.490	**0.985**	−0.040	**0.951**	−0.186
Coagulation—colorimetric models						
RCT	−0.255	**0.630**	0.028	**−0.820**	0.191	**−0.630**
k20	0.227	**0.735**	0.041	**−0.693**	−0.119	**−0.712**
A60	−0.493	−0.426	0.138	**0.793**	0.324	−0.401
CY	**−0.966**	0.096	**−0.852**	0.153	**0.840**	−0.080
DCY	0.314	**−0.501**	−0.361	0.026	0.265	0.271
L*	**−0.889**	−0.104	**−0.673**	**0.590**	**0.526**	**0.836**
a*	−0.187	**0.982**	**−0.821**	−0.120	**0.735**	0.272
b*	**−0.851**	0.150	**−0.983**	0.185	**0.953**	−0.212

* FAT = fat content; CP = crude protein; LAC = lactose content; RCT = rennet clotting time; k_20_ = curd firming time; A_60_ = curd firmness at 60 min; CY = curd yield; DCY = dry curd yield; L* = lightness; a* = red/green balance; b* = blue/yellow balance.

## Data Availability

Data is contained within the article.

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
