# Peer review of "Exploring Interrelationships between Colour, Composition, and Coagulation Traits of Milk from Cows, Goats, and Sheep"

_foods, 2024, doi:10.3390/foods13040610_

Round 1

Reviewer 1 Report

Comments and Suggestions for Authors

This is a review of the manuscript ''Exploring interrelationships between colour, composition, and coagulation traits of milk from cows, goats and sheep''

The issues below will need to be addressed.

Check that the abstract is about 200 words as per the guide.

Sample collection: The text says samples were 10-year historic samples.  Over 10 years, were samples analysed with the same equipment?

Line 72: Authors should briefly describe how the random selection was carried out. What were the criteria used?

Line 77: What was the temperature of the cold storage?

Line 134 .... Text shows Error! Reference source not found. presents the..... Revise. The phrase was repeated in lines  162, 165,168,192, 204, 208, 227, 236, 239, 248, 249. The meaning of some sentences was lost. Check the rest of the manuscript. The number of times this occurred suggests a lack of proper proofreading or could it be from the editorial software?

References 19 and 20 were used for statistical analysis but how the analysis was carried out is not clear. Authors should provide the page numbers and mention briefly in the methods section why 'Canonical' analysis was used.

Nothing was mentioned about the link between the colour and taste of the milk in the discussion. This can be added as a future study. 

The conclusion in Line 315 is redundant because the authors mentioned the same thing in reference 12.

Overall authors re-emphasised the importance of colour as per previous publications. 

Comments on the Quality of English Language

The issues above need to be addressed.

Author Response

This is a review of the manuscript ''Exploring interrelationships between colour, composition, and coagulation traits of milk from cows, goats and sheep''

The issues below will need to be addressed.

AU: We kindly appreciate the time and effort of the reviewers in evaluating this manuscript. We found their comments very useful, and we hope we have addressed them all in this improved new version of the manuscript. Please find the detailed responses below and the corresponding revisions/corrections highlighted in yellow in the re-submitted files. Thank you.

Check that the abstract is about 200 words as per the guide.

AU: Thank you for pointing this out. We have revised the abstract and reduced it to no more than 200 words, as suggested in the Instructions for Authors.

Sample collection: The text says samples were 10-year historic samples.  Over 10 years, were samples analysed with the same equipment?

AU: Thanks for the question. Yes, samples were always analysed using the same equipment and, this is the main reason why we only used samples from the last 10 years. This equipment includes a Formagraph lactodinamograph for coagulation performance and a Milkoscan FT120 for analysis of milk composition (fat, protein, lactose, solids), both from Foss Electric (Hillerød, Denmark). The color meter and pH meter used were also the same for all samples (PCE-CSM2 and Crison Basic20, respectively).

Line 72: Authors should briefly describe how the random selection was carried out. What were the criteria used?

AU: A sentence has now been included in the Materials and Methods section to describe the random selection. Selection of samples was carried out in MS Excel (Microsoft Corp., Redmond, WA) using the RANDBETWEEN function. Briefly, a number was assigned to each database record for each species starting with 1. The RANDBETWEEN function was then used to generate 800 random numbers between 1 and the maximum for each data set. Then the records corresponding to the randomly generated numbers were selected. After each random number was generated, it was checked if that number had already been selected previously. If the randomly generated number was already present in the list of previously selected numbers, another random number was generated again until a number that was not repeated was obtained.

Line 77: What was the temperature of the cold storage?

AU: Samples were kept at 4ºC until analysis. This has now been included in the Material and Methods section of the manuscript.

Line 134 .... Text shows Error! Reference source not found. presents the..... Revise. The phrase was repeated in lines 162, 165,168,192, 204, 208, 227, 236, 239, 248, 249. The meaning of some sentences was lost. Check the rest of the manuscript. The number of times this occurred suggests a lack of proper proofreading or could it be from the editorial software?

AU: Our sincere apologies for this. All 6 authors have done a thorough proofreading of the manuscript before submission and no issues were spotted. Tables and figures were cited in the text and linked using the cross-reference tool from MS Word. However, for some reason, formatting has been altered when uploading the file to MDPI submission system. We have now removed these links and input the references in plain text to avoid this issue when submitting the reviewed document. Table and figure names should hopefully now read correctly.

References 19 and 20 were used for statistical analysis but how the analysis was carried out is not clear. Authors should provide the page numbers and mention briefly in the methods section why 'Canonical' analysis was used.

AU: Following suggestions, the page numbers of the cited books have been added to each reference in the references section. In addition, in order to clarify the use of canonical analysis, the following paragraph has been added to the Statistical Analysis subsection (L106-109): "Canonical discriminant analysis was chosen for its effectiveness in reducing dimensionality and optimizing the separation between groups, offering a clear interpretation of the contributions of variables to discrimination, especially in large datasets".

Nothing was mentioned about the link between the colour and taste of the milk in the discussion. This can be added as a future study. 

AU: Done. A new sentence has been included in the discussion to introduce this relationship and the importance of considering it for future studies. This is also supported by the inclusion of 2 new literature references.

The conclusion in Line 315 is redundant because the authors mentioned the same thing in reference 12.

AU: We agree with the reviewer and, following the suggestion, we have removed one of the sentences to avoid redundancy.

Overall authors re-emphasised the importance of colour as per previous publications. 

AU: We appreciate the comment. Yes, the importance of colour in the assessment of milk has been stressed, although little work has been done so far on this aspect in small ruminants. However, the highlight of the present study is the use of colour traits as a potential rapid and inexpensive tool to discriminate between milk from the three most common domestic ruminant species.

Reviewer 2 Report

Comments and Suggestions for Authors

The present study highlights particularities in composition, colour, and coagulation properties of milk from cow, sheep and goat, proving sheep milk stands out for its superior cheesemaking capacity. This is good job and would be potentially valuable for dairy industry. However, due to formatting issues and Figures and Tables are not clearly indicated in main text, I would suggest the authors check and revise the manuscript thoroughly and clearly before resubmitting.

Comments:

It is recommended to add 1-2 sentences about the use of coagulation characteristics in production, e.g., the authors state that many studies compare the coagulation characteristics of different milks (Lines 41-47), but it is not clear to the general reader what the significance of coagulation is in terms of its application in production.

All Tables and Figures were not stated clearly in the main text. When it describes the results of milk characteristic and differences, I didn’t know which table or figures I should move to and read.  

L98. …presented in Error!?

L134. ‘Error! Reference source not found.Presents the outcomes of ……same for L162, L165, L168, L181, L190, L205, L208, L220, Etc., please check through the manuscript and revise them.

L136. ‘The table emphasizes ……’. So which table does it mean, Table 1 or Table 2. Please show all Tables and figures specifically, like Table 1, Figure 2. Etc.  

Author Response

The present study highlights particularities in composition, colour, and coagulation properties of milk from cow, sheep and goat, proving sheep milk stands out for its superior cheesemaking capacity. This is good job and would be potentially valuable for dairy industry. However, due to formatting issues and Figures and Tables are not clearly indicated in main text, I would suggest the authors check and revise the manuscript thoroughly and clearly before resubmitting.

AU: We kindly appreciate the time and effort of the reviewers in evaluating this manuscript. We found their comments very useful, and we hope we have addressed them all in this improved new version of the manuscript. Please find the detailed responses below and the corresponding revisions/corrections highlighted in yellow in the re-submitted files. Thank you.

Comments:

It is recommended to add 1-2 sentences about the use of coagulation characteristics in production, e.g., the authors state that many studies compare the coagulation characteristics of different milks (Lines 41-47), but it is not clear to the general reader what the significance of coagulation is in terms of its application in production.

AU: Thank you for this comment. We fully agree on this and following the suggestion we have added a couple of sentences in this section to further introduce the concept of coagulation performance and its significance for cheesemaking and dairy production. We have also included two new literature references to support this.

All Tables and Figures were not stated clearly in the main text. When it describes the results of milk characteristic and differences, I didnt know which table or figures I should move to and read.

AU: Our sincere apologies for this. All 6 authors have done a thorough proofreading of the manuscript before submission and no issues were spotted. Tables and figures were cited in the text and linked using the cross-reference tool from MS Word. However, for some reason, formatting has been altered when uploading the file to MDPI submission system. We have now removed these links and named the references in plain text to avoid this issue when submitting the reviewed document. Table and figure names should hopefully read now correctly.

L98. …presented in Error!?

AU: Please see answer to previous comment.

L134. Error! Reference source not found.Presents the outcomes of ……same for L162, L165, L168, L181, L190, L205, L208, L220, Etc., please check through the manuscript and revise them.

AU: Please see answer to previous comment.

L136. The table emphasizes ……’. So which table does it mean, Table 1 or Table 2. Please show all Tables and figures specifically, like Table 1, Figure 2. Etc.  

AU: Please see answer to previous comment.